# Identification of the co-regulatory siRNAs of "miRNA→target" in *Oryza sativa*

Zhihong Yang[1], Lan Yu[1], Yeqing Jiang[1], Yijun Meng[2]*, Chaogang Shao[1]*

**1** College of Life Sciences, Huzhou University, Huzhou, Zhejiang, P.R. China, **2** College of Life and Environmental Sciences, Hangzhou Normal University, Hangzhou, Zhejiang, P.R. China

* mengyijun@zju.edu.cn (YM); shaocg@zjhu.edu.cn (CS)

## Abstract

The current "small interfering RNA(siRNA)→Target" mining tools can only search for targets of known siRNAs, and cannot discover co-regulatory siRNAs of unknown sequences that may exist, which means that the "microRNA(miRNA)→Target" database obtained by these mining tools is incomplete. Using the previously developed sRNATargetDigger, we re-mined the rice "miRNA→Target" database supported by the degradome and found 86.2% of the target genes were co-regulated by one or more miRNAs\siRNAs. Besides the known miRNAs, 30 miRNA isoforms (isomiRs) and 12 siRNAs were identified to be involved in co-regulation, which play important roles in rice response to external auxin regulation, rice blast resistance, adventitious root formation, cold resistance, and tillering etc. Some isomiRs even have higher expression levels than miRNAs. In addition, we also found that the regulatory relationship between 51 miRNAs and 48 target genes in the original database could not be verified due to the low expression levels of miRNA, poor complementarity between miRNA and target, or no specific cleavage signal detected by degradome in the middle of the miRNA binding site in the targets. Four miRNAs (osa-miR530-5p,osa-miR319b,osa-miR172c and osa-miR395a) only found isomiRs involved in regulation. In addition, we also found a number of miRNA→target regulatory relationships missed in the database. This study improved the rice "miRNA→target" database which will contribute to the research of rice miRNA and molecular breeding.

## Introduction

MicroRNA (miRNA) is a class of evolutionarily conserved, endogenous single-stranded small RNA(sRNA) with a length of about 19-24 nucleotides. It can cause the cleavage and degradation or translation inhibition of target genes through specific base complementary pairing with target mRNAs, and affecting almost all biological processes from development, physiology to stress response. It is a key factor in post-transcriptional gene expression regulation [1,2]. Recent studies have shown that trans-acting small interfering RNA(ta-siRNA), natural antisense small interfering RNA(nat-siRNA), coding transcript-derived small interfering RNA(ct-siRNA) and other sources of short interfering RNA (siRNA) also have similar gene regulatory functions as miRNA, and play an important role in plant growth and development [3].

**Data availability statement:** The sRNA HTS data sets (GSM1081563, GSM1081564,

GSM1081565, GSM1409671, GSM1409672, GSM1409673, GSM1409674, GSM1547539, GSM816687, GSM816688, GSM816689, GSM816690, GSM816691, GSM816692, GSM816693, GSM816694, GSM816695, GSM816696, GSM816697, GSM816698, GSM816699, GSM647192, GSM647193, GSM647194, GSM647195, GSM816730, GSM816731, GSM816732, GSM816733, GSM816735, GSM816736, GSM816737, GSM816738, GSM816739, GSM816740, GSM816741, GSM816742, GSM816743, GSM816744) and the degradome sequencing data sets (GSM434596, GSM455938, GSM4113551, GSM455939, GSM476257) of Oryza sativa were retrieved from GEO (Gene Expression Omnibus; http://www.ncbi.nlm.nih.gov/geo/). The cDNAs and the gene annotations of Oryza sativa were retrieved from the FTP site of the rice genome annotation project (Release7; https://rice.uga.edu/pub/data/Eukaryotic_Projects/o_sativa/annotation_dbs/pseudomolecules/version_7.0/). The miRNA-target database of Oryza sativa were retrieved from the TarDB (http://www.biosequencing.cn/TarDB/download.html) and the miRNAs of Oryza sativa were retrieved from miRBase (release 22.1; https://www.mirbase.org/). The mining software, sRNATargetDigger, can be downloaded from the Project sRNA-TargetDigger in OSF database (https://osf.io/n7uc9/files/osfstorage).

**Funding:** This work was supported by the National Natural Sciences Foundation of China[31771457].

**Competing interests:** The authors have declared that no competing interests exist.

Identifying these siRNAs target genes is an important step in our understanding of their functions. Bioinformatics technology can help predict the target genes of siRNAs. Since plant miRNAs mainly bind to target sites through near-perfect complementary pairing, which leads to direct cleavage of target mRNAs, the pattern of plant miRNA recognizing target sites is relatively simple [4]. However, whether the predicted target genes are accurate generally needs biological verification. The commonly used methods are as follows: one is the Agrobacterium injection method, in which the constructed vector containing miRNA and target genes is injected into tobacco leaves to detect the cleavage results of miRNA on target genes [5], the second is to use wheat germ lysate to detect the cleavage activity of miRNA on target genes in vitro [6], and the third is to directly analyze the products of miRNA cleavage using 5'rapid-amplification of cDNA ends (5'RACE). This method involves attaching a linker to the 5' end of the miRNA cleavage product, and after Reverse Transcription-Polymerase Chain Reaction(RT-PCR) and sequencing, it can not only verify the cleavage characteristics of miRNA on target genes, but also identify the precise cleavage sites of miRNA. It is currently the most reliable method for identifying miRNA target genes [5]. However, the above experimental verification methods are relatively cumbersome, which greatly limits the functional identification of miRNA.

High-throughput sequencing (HTS) technology is a breakthrough sequencing technology developed in recent years, which can determine millions of nucleic acid sequences at a time. Depending on the RNA separation conditions, HTS can be used to determine the transcriptome of mRNAs, the degradome of mRNAs, and the transcriptome of sRNA. It can even obtain various sRNAs loaded in the Argonaute (AGO) complex through specific antibody capture technology, which provides a powerful tool for the identification of miRNAs and their target genes [7–9]. In particular, the degradome high-throughput sequencing technology [10], or named as parallel analysis of RNA ends (PARE)) [11] and nanoPARE techniques [12], can capture all the degradation fragments of transcripts, and by using bioinformatics tools (CleaveLand [13], SeqTar [14], PAREsnip2 [15], sPARTA [16], etc.) to match with reference sequences, it can detect specific cleavage signals on transcripts, which can provide experimental evidence for miRNAs to perform cutting functions, and can be used for large-scale identification of the targets of plant miRNAs [10–12].

Due to the regulatory relationship between miRNAs and target genes being a "multi" to "multi" regulatory network, a target gene may be co-regulated by multiple miRNAs or siRNAs at the same or different sites. However, current "siRNAs→Target" mining tools based on degradome can only search for target genes of known sequence siRNAs, and cannot discover co-regulatory siRNAs of unknown sequences that may exist. Therefore, the miRNA→target regulatory network obtained by these mining tools may be incomplete.Besides, according to the previous reports, the effectiveness of siRNA in cleaving and inhibiting mRNA is positively correlated with the concentration of siRNA [17].So, if there are unknown co-regulatory siRNAs with much higher expression levels than known miRNAs, the contribution of this miRNA to target gene cleavage is very little, and the previously reported miRNA→target regulatory relationship may be a false positive result.

To address this issue, we integrated reverse mining technology with the classic "siRNA→Target" mining algorithm to develop a new mining tool, sRNATargetDigger [18]. This tool can search for all possible co-regulatory siRNAs from HTS data of sRNAs, and determine the degree of participation of these siRNAs in target regulation based on their expression levels, thereby obtaining a more complete and reliable "miRNAs\siRNA→Target" regulatory relationship and also enabling bidirectional mining [18]. Testing

of "miRNAs→Target" in *Arabidopsis thaliana* showed that 69 miRNA target genes found co-regulatory siRNAs in one or more tissues, forming 170 new siRNA→target regulatory relationships [18].

Rice is an important food crop and a model plant for life science research. The "miRNA→Target" database is of great significance for basic research and molecular breeding of rice. In this study, we used the sRNATargetDigger tool and massive HTS data from public databases to re-mine the reported rice "miRNA→Target" database, searching for missing co-regulatory siRNAs information.

## Materials and methods

### Data sources

The cDNAs and the gene annotations of *Oryza sativa* (Nipponbare) were retrieved from the FTP site of the rice genome annotation project (Release 7; https://rice.uga.edu/pub/data/Eukaryotic_Projects/o_sativa/annotation_dbs/pseudomolecules/version_7.0/) [19]. The sRNA HTS data sets (GSM1081563, GSM1081564, GSM1081565, GSM1409671, GSM1409672, GSM1409673, GSM1409674, GSM1547539, GSM816687, GSM816688, GSM816689, GSM816690, GSM816691, GSM816692, GSM816693, GSM816694, GSM816695, GSM816696, GSM816697, GSM816698, GSM816699, GSM647192, GSM647193, GSM647194, GSM647195, GSM816730, GSM816731, GSM816732, GSM816733, GSM816735, GSM816736, GSM816737, GSM816738, GSM816739, GSM816740, GSM816741, GSM816742, GSM816743, GSM816744) and the degradome sequencing data sets (GSM434596, GSM455938, GSM4113551, GSM455939, GSM476257) of *Oryza sativa* were retrieved from GEO (Gene Expression Omnibus; http://www.ncbi.nlm.nih.gov/geo/) [20]. See S1 Table for the sequencing library information of these data sets. In order to allow cross-library comparison, the normalized read count (in RPM, reads per million) of a short read from a specific library was calculated by dividing the raw count of this read by the total counts of the library, and then multiplied by $10^6$.

### Prediction of miRNA→target binding sites and identification of specific cleavage signals in target genes

We downloaded the newly published miRNA→target database of *Oryza sativa* from the TarDB (http://www.biosequencing.cn/TarDB/download.html) [21], and extracted the sequences of miRNAs and targets from the miRBase (release 22.1, https://www.mirbase.org/) [22] and rice cDNA databases(Release 7; https://rice.uga.edu/pub/data/Eukaryotic_Projects/o_sativa/annotation_dbs/pseudomolecules/version_7.0/) [19], respectively.Then the binding sites of miRNA on target were predicted by miRU algorithm [23] to distinguish two categories of co-regulatory siRNAs which have the same or different binding site with miRNAs on target genes. Next, we matched the degradome to the target, recorded the 5'-end site information of the exact match (if there are multiple exact matches, they all need to be recorded) and calculated the degradation signal intensity of cleavage site and background as previously reported [18]:

Degradation signal intensity of cleavage site = sum of the expression counts of all matching degradome sequences at the cleavage site/sum of the number of matching degradome sequences at the cleavage site;

Degradation signal intensity of background = sum of the expression counts of the matching degradome sequences in full-length (except for the cleavage site)/sum of the number of matching degradome sequences in full-length(except for the cleavage site).

Screening of specific cleavage sites: The degradation signal intensity of the cleavage site should be 5 times or more of the background degradation signal intensity.

## Co-regulatory siRNA identification

The samples were divided into two different tissues, seedling and panicle, and the mining was conducted separately. There were a total of 21 sets of sRNA sequencing data and 3 degradome data for seedling, and 18 sets of sRNA sequencing data and 2 degradome data for panicle (S1 Table).

(1)  Search for co-regulatory siRNAs with the same cleavage site as miRNA: Using the forward mining module of sRNATargetDigger (https://osf.io/n7uc9/files/osfstorage), input the degradation group data, sRNA sequencing data, rice miRNA and target gene sequences, and search co-regulatory siRNAs with the same cleavage site as miRNA. The expression counts of co-regulatory siRNAs needs to be greater than or equal to 10 rpm, and not less than 1/10 of the highest value of the regulatory miRNA/siRNA at that site.

(2)  Search for co-regulatory siRNAs with different cleavage sites from miRNA: Utilize the reverse mining module of sRNATargetDigger, input specific cleavage site information which is not in the middle of miRNA binding site, sRNA sequencing data, and target gene sequence to search co-regulatory siRNAs. The expression value of co-regulatory siRNA needs to be greater than or equal to 10 rpm, and not less than 1/10 of the highest value of the regulatory siRNA at that site.

The co-regulatory siRNAs obtained will be annotated as miRNA, isomiR or other siRNA by searching the miRNA and pre-miRNA databases.

## Screening and verification of the results

The co-regulatory siRNAs discovered by sRNATargetDigger were further screened and validated by psRNATarget (https://www.zhaolab.org/psRNATarget/) [24], a plant sRNA target analysis server. "Submit small RNAs and Target" module of psRNATarget and the default parameters were selected. Only the predicted results with "Inhibition: Cleavage" were retained, and the cleavage signal found by degradome on the target should be located in the middle of the binding site of the co-regulatory siRNA.

For miRNA→target in the TarDB database [21] that failed to pass the sRNATargetDigger, we further analyzed the expression of the miRNA, the cleavage signal on the target, and the binding of miRNA to the target for verification. If the miRNA is not expressed or expressed very low (<10rpm), or if no cleavage signal is detected by the degradome in the middle of the binding site of the miRNA on the target, or if both the miRU and psRNATarget detections indicate that the miRNA cannot bind to the target, it is determined that the miRNA→target regulatory relationship is incorrect in these tissues or conditions.

## Results and discussion

### Discovery of anomalies in the rice "miRNA→target" database

We downloaded 480 miRNA→target pairs of rice supported by the degradome from the newly published TarDB database [21], including 98 miRNAs and 165 target genes, for re-mining. The regulatory relationships between 62 miRNAs and 116 genes were validated by sRNA-TargetDigger (S2 and S3 Tables). The rest which could not pass the verification were further analyzed. miRU and psRNATarget found osa-miR408-5p, osa-miR5541 and osa-miR2098-3p could not bind with their corresponding targets, LOC_Os04g39350.1 (OsHIPP19, heavy-metal-associated isoprenylated plant protein), LOC_Os09g32700.1(expressed protein) and LOC_Os09g39750.1(domain of unknown function DUF966 domain containing protein), respectively (S4 Table) .Forty one target genes did not find cleavage signals in the middle

of the miRNA binding sites by degradome. ([Fig 1](), [S1 Fig](), [S4 Table]()) and expression analysis found 22 miRNAs which regulate 44 targets were almost not expressed in all samples (S4 Table). Considering that the sRNA data used by sRNATargetDigger for mining is exactly the same as that provided by the TarDB database, and the degradome data have one more group (GSM476257) than that provided by the TarDB (theoretically, more cleavage signals can be detected, improving the success rate of miRNA→target prediction), the impact of the datasets on the results can be ruled out. So, the regulation relationship between these miRNA→target pairs that could not be validated by sRNATargetDigger may not be correct in these tissues or conditions. Due to the limitation of samples, we cannot rule out the possibility that the regulatory relationship may be exist in other tissues or under specific conditions.

In addition, we found that many miRNAs belonging to the same family of rice in the TarDB database share similar sequences and co-regulate the same target gene. However,

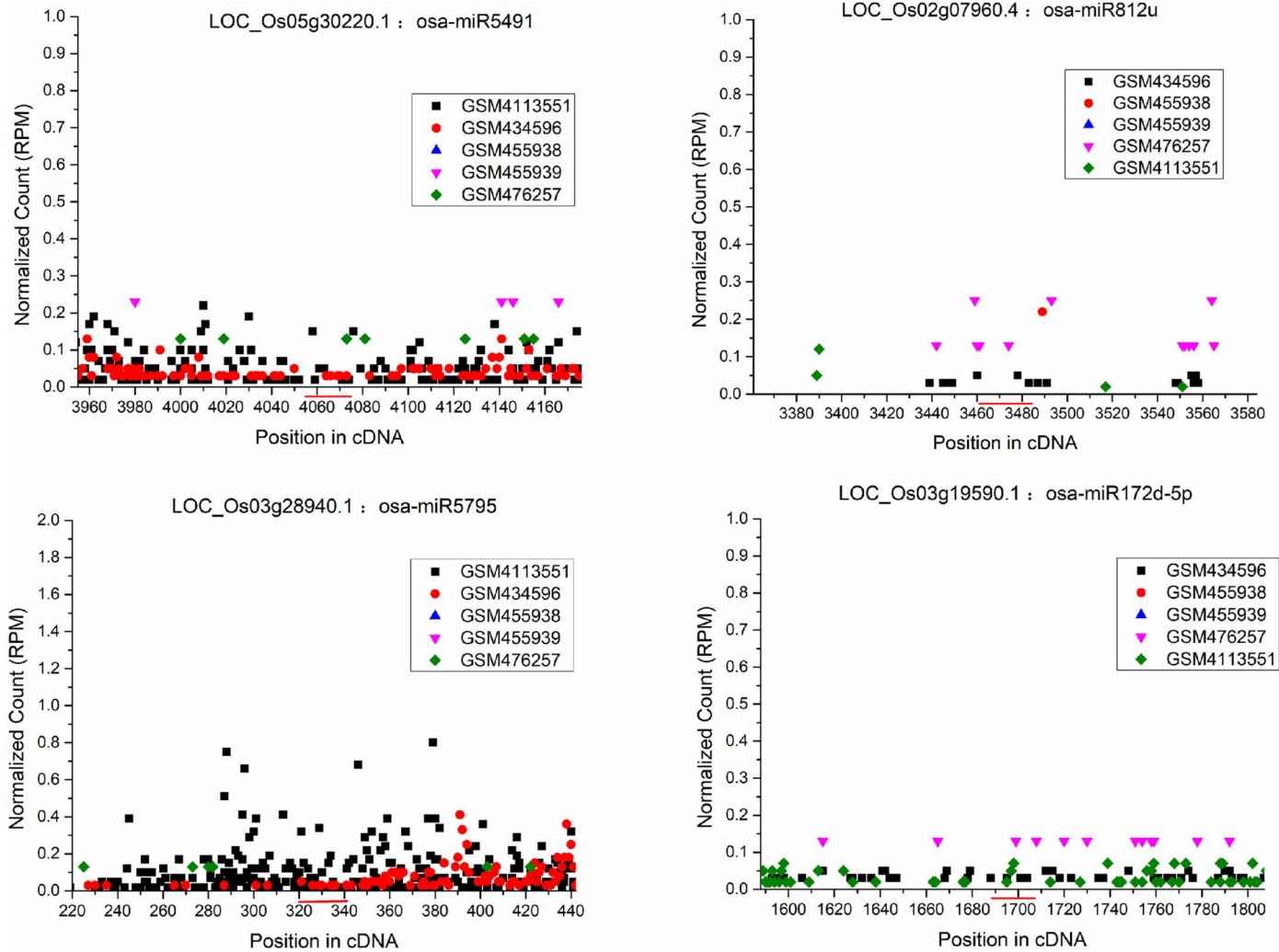

**Fig 1. Four examples of the target plots (t-plots) which did not find cleavage signals in the middle of the miRNA binding sites by degradome.** All 5 degradome sequencing data libraries (GSM434596, GSM455938, GSM4113551, GSM455939, GSM476257) were recruited for T-plot profiling. The IDs of the target transcripts and the corresponding miRNAs are listed on the top of sub-figures. The *x* axes measure the positions of the degradome signals along the transcripts, and the *y* axes measure the degradome signal intensities based on normalized counts (in RPM, reads per million).The binding sites of the miRNA on their target transcripts were denoted by red horizontal lines. No specific cleavage signals are found in the middle of the miRNA binding sites in these target genes.

in the co-regulation relationship, the TarDB database does not provide information on the contribution degree of each miRNA. Through sRNATargetDigger mining, we found that there are significant differences in the expression levels of different members of many miRNA families. For example, three members of the osa-miR164 family (osa-miR164d,osa-miR164e and osa-miR164f) co-regulate LOC_Os06g46270.1(OMTN4, no apical meristem protein) in rice seedlings, but high-throughput sequencing shows the abundant of osa-miR164f is more than four times than that of osa-miR164d and osa-miR164e in various datasets, and even more than 100 times in some datasets. According to the previous reports, the effectiveness of siRNA in cleaving and inhibiting mRNA is positively correlated with the concentration of siRNA [17]. So, osa-miR164f is the main regulator of LOC_Os06g46270.1 (Fig 2A). In rice panicle, osa-miR396c-5p and osa-miR396e-5p co-regulate LOC_Os02g45570.2(OsGRF10, growth regulating factor), but the expression level of osa-miR396e-5p in various datasets also far exceeds that of osa-miR396c-5p, making it the main regulator of LOC_Os02g45570.2 (Fig 2B). Besides, we found 6 miRNAs in 5 miRNA families were expressed at less than 1/10 of the highest expression miRNA which cut the target. In theory, their contribution to target gene cleavage is very little (S4 Table).

During the re-mining, sRNATargetDigger also found some missing miRNA→target regulatory relationships in the TarDB database (Table 1).For example, in the original database, osa-miR444c.2 only regulates LOC_Os02g49840.3(OsMADS57, MADS-box gene) and LOC_Os02g49840.4 (OsMADS57), but in rice seedlings, we found that osa-miR444c.2 also participates in the regulation of LOC_Os02g49840.2 (OsMADS57) (S2 Table). In addition to

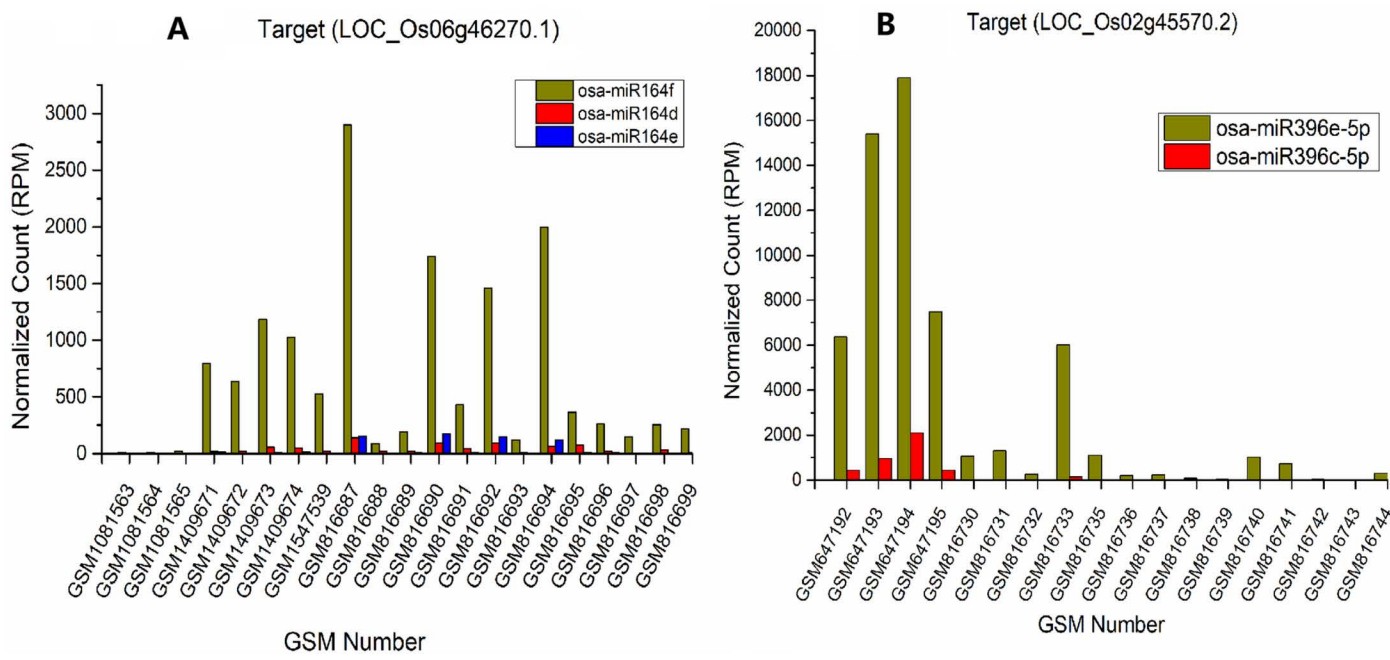

**Fig 2. Expression bar graph of the miRNAs belonging to the same family involved in the co-regulation of target genes. A.** osa-miR164d,osa-miR164e and osa-miR164f co-regulate LOC_Os06g46270.1 in rice seedlings. The *x* axes represents the GSM numbers of 21 samples in seedling and the *y* axes represents the expression values of miRNAs (in RPM, reads per million).The abundant of osa-miR164f is more than that of osa-miR164d and osa-miR164e in various datasets, and it is obviously the main regulator of LOC_Os06g46270.1(OMTN4). **B.** osa-miR396c-5p and osa-miR396e-5p co-regulate LOC_Os02g45570.2 (OsGRF10, growth regulating factor) in rice panicle.The *x* axes represents the GSM numbers of 18 samples in panicle and the *y* axes represents the expression values of miR-NAs (in RPM, reads per million).The expression level of osa-miR396e-5p in various datasets far exceeds that of osa-miR396c-5p, making it the main regulator of LOC_Os02g45570.2.

**Table 1. Missing or novel miRNA→target regulatory relationships discovered by sRNATargetDigger in rice.**

| miRNAs | Targets | Samples |
|---|---|---|
| osa-miR444e | LOC_Os02g36924.1 (OsMADS27) | Seedling, Panicle |
| osa-miR444c.1 | LOC_Os02g36924.1(OsMADS27) | Seedling |
| osa-miR444e | LOC_Os04g38780.1(transcription factor) | Seedling |
| osa-miR444c.2 | LOC_Os02g49840.2 (OsMADS57) | Seedling |
| osa-miR169o | LOC_Os03g07880.1(OsNF-YA1), LOC_Os03g07880.2(OsNF-YA1), LOC_Os03g07880.3 (OsNF-YA1), LOC_Os03g29760.1(OsNF-YA2), LOC_Os03g29760.2(OsNF-YA2), LOC_Os03g29760.3(OsNF-YA2), LOC_Os07g41720.1(OsNF-YA6), LOC_Os07g41720.2(OsNF-YA6) | Seedling, Panicle |
| osa-miR169o | LOC_Os07g06470.1(OsNF-YA5), LOC_Os07g06470.2(OsNF-YA5) | Seedling, |
| osa-miR169a, osa-miR169c | LOC_Os03g44540.1(OsNF-YA3), LOC_Os03g48970.1(OsNF-YA4), LOC_Os03g48970.2 (OsNF-YA4), LOC_Os03g48970.3(OsNF-YA4), LOC_Os03g48970.4(OsNF-YA4) | Seedling, Panicle |
| osa-miR169e | LOC_Os03g44540.1(OsNF-YA3), LOC_Os03g48970.1(OsNF-YA4), LOC_Os03g48970.2 (OsNF-YA4), LOC_Os03g48970.3(OsNF-YA4), LOC_Os03g48970.4(OsNF-YA4) | Panicle |
| osa-miR166k-3p | LOC_Os04g48290.1(MATE efflux family protein) | seedling |

being regulated by osa-miR444c.2, LOC_Os02g36924.1(OsMADS27, MADS-box family gene) is also found to be co-regulated by osa-miR444e at same site (cutsite 575) and osa-miR444c.1 at different site (cutsite 569) (Fig 3, S2 Table).All these missing miRNA→target pairs identified by sRNATargetDigger passed the psRNATarget validation (S5 Table). We further investigated the relevant literature on the mining of miRNA target genes in rice and found our partial results, the regulatory relationships of "osa-miR444c.1→LOC_Os02g36924.1 (OsMADS27)", "osa-miR444c.2→LOC_Os02g49840.2(OsMADS57)", "osa-miR444e→LOC_Os02g36924.1 (OsMADS27)" and "osa-miR444e→ LOC_Os04g38780.1(transcription factor)" were also supported by the research of Fei *et al.* [25].

## Co-regulatory siRNAs identified by sRNATargetDigger in rice

Among the 116 target genes verified by sRNATargetDigger in rice seedlings and panicles, 100 target genes were found to be co-regulated by two or more miRNAs\siRNAs (S2 and S3 Tables). Most of the co-regulation occurred at the same site except for 5 targets. Only 16 target genes were regulated by one miRNA or isomiR. Among them, four miRNAs, namely osa-miR530-5p, osa-miR319b, osa-miR172c and osa-miR395a, have low expression levels and sRNATargetDigger only found corresponding isomiRs involved in the regulation (S2 and S3 Tables).

IsomiRs are predominantly derived from the alternative and imprecise cleavage of Drosha and Dicer during pri-miRNA/pre-miRNA processing, 3' addition events in miRNA maturation processes or base modification of pri-miRNA/pre-miRNA/miRNA [26]. They are very close to miRNA sequences and RNAi machinery generates isomiRs as a potential strategy to increase the specificity and efficiency of silencing target genes [26]. In this study, sRNATargetDigger identified 30 isomiRs which play an important role in co-regulation of the targets in rice (S2 and S3 Tables) and these newly discovered isomiR→target regulatory relationships were further confirmed by psRNATarget (S6 Table).Some isomiRs have very high expression

levels, even exceeding miRNAs. For example, osa-miR167d-5p, which plays an important role in rice response to external auxin regulation [27] and rice blast resistance [28], has been found co-regulation with osa-miR167c-5p and isomiR of osa-miR167d-5p to target ARF8 (LOC_Os04g57610) in seedling and panicle (S2 and S3 Tables). In most sRNA sequencing data, the expression level of isomiR of osa-miR167d-5p far exceeds that of osa-miR167d-5p and osa-miR167c-5p (Fig 4).Although literature reported that 22-nt miRNAs had the ability to trigger the phased siRNAs production from the cleaved targets [29], further analysis did not find that this 22-nt isomiR of osa-miR167d-5p can stimulate ARF8 to produce phased siRNAs. In Arabidopsis, Li *et al.* also found that many MIR genes produce 22-nt isoforms due to imprecision in DCL1 processing, yet most 22-nt miRNA isoforms do not trigger phasiRNA production from their target RNAs [30].The target gene OsDRM2 (LOC_Os03g02010.4) of osa-miR820c play an important role in regulating rice vegetative and reproductive growth through DNA

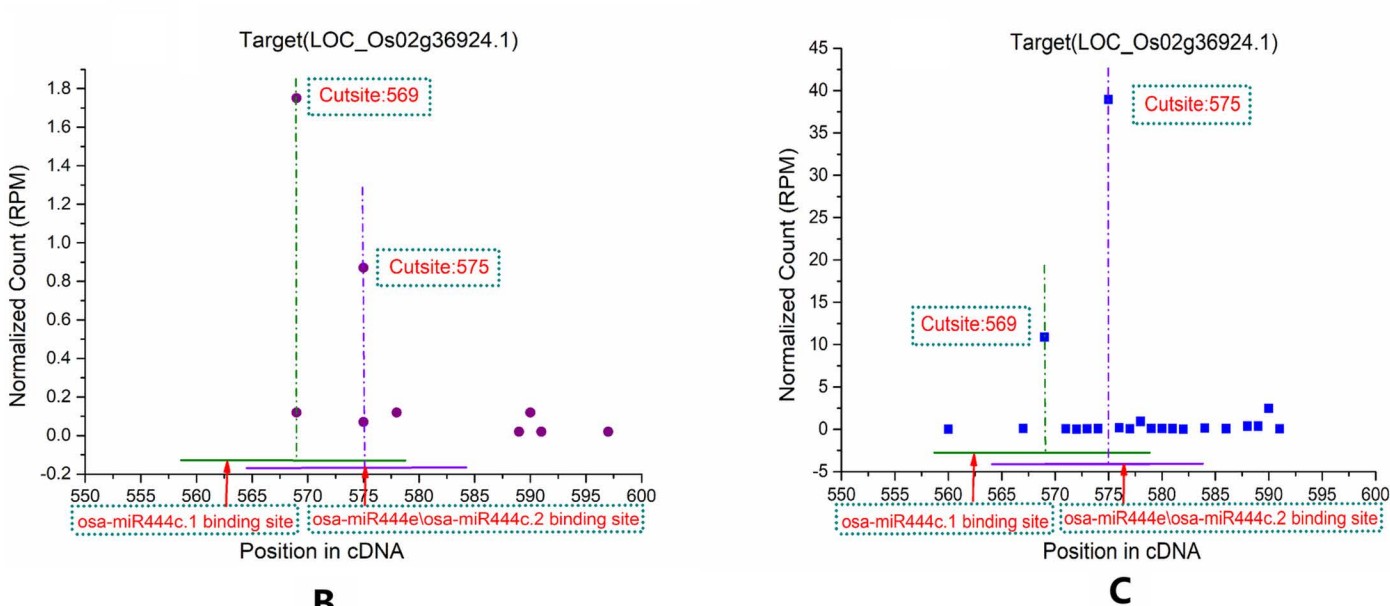

**Fig 3. osa-miR444c.2 and co-regulating miRNAs with their target LOC_Os02g36924.1 in rice seedling.** A. Expression of osa-miR444c.2, osa-miR444e and osa-miR444c.1 in the samples of wt2003, wt2007 and wt2011, and their alignments with target genes. B and C. Degradome sequencing data libraries GSM4113551 (B) and GSM434596 (C) were recruited for T-plot profiling, respectively. The ID of the target LOC_Os02g36924.1(OsMADS27) is listed on the top of the figure. The y axes measure the normalized reads (in RMP, reads per million) of the degradome signals, and the x axes represent the position of the cleavage signals on the target transcripts. The binding site of the osa-miR444e\osa-miR444c.2 and osa-miR444c.1 on their target was denoted by purple and green horizontal line, respectively. The vertical dotted line indicates the specific cleavage site.

methylation [31]. sRNATargetDigger found that there are five other isomiRs of osa-miR820c and four siRNAs involved in the co-regulation.One of the isomiRs of osa-miR820c,"UCGG CCUCGUGGAUGGACCAGGAG", has a higher expression level than osa-miR820c in most samples (S2 Table).

The main root of rice will die after one week of germination, and the adventitious roots that grow later are the main organs for rice to fix, absorb nutrients and water, affecting the growth, yield and resistance of rice. Highly expressed osa-miR156j-5p regulates the growth hormone transport and signaling by inhibiting the activation of OsMADS50 by OsSPL3 (LOC_Os02g04680)/OsSPL12 (LOC_Os06g49010), promoting the occurrence of adventitious

**A**

| sRNA-ID | sRNA-count (rpm) | | | | | sRNA-Target Alignment |
|---|---|---|---|---|---|---|
| | Control (GSM816687) | Drought (GSM816690) | Salt (GSM816692) | Cold (GSM816694) | Heat (GSM816696) | |
| osa-miR167d-5p | 2402.36 | 1846.27 | 1802.71 | 1389.64 | 28.42 | sRNA 21 AUCUAGUACGACCGUCGAAGU 1<br> ::::::: ::::::::::::<br>Target 2658 UAGAUCAGGCUGGCAGCUUGU 2678 |
| osa-miR167c-5p | 3499.03 | 2888.27 | 2961.64 | 1731.36 | 133.98 | sRNA 21 GUCUAGUACGACCGUCGAAGU 1<br> .::::::: ::::::::::::<br>Target 2658 UAGAUCAGGCUGGCAGCUUGU 2678 |
| isomiR of osa-miR167d-5p | **16879.51** | **15243.31** | **13413.32** | **10970.09** | **312.63** | sRNA 22 AGUCUAGUACGACCGUCGAAGU 1<br> .::::::: ::::::::::::<br>Target 2657 AUAGAUCAGGCUGGCAGCUUGU 2678 |

**B**

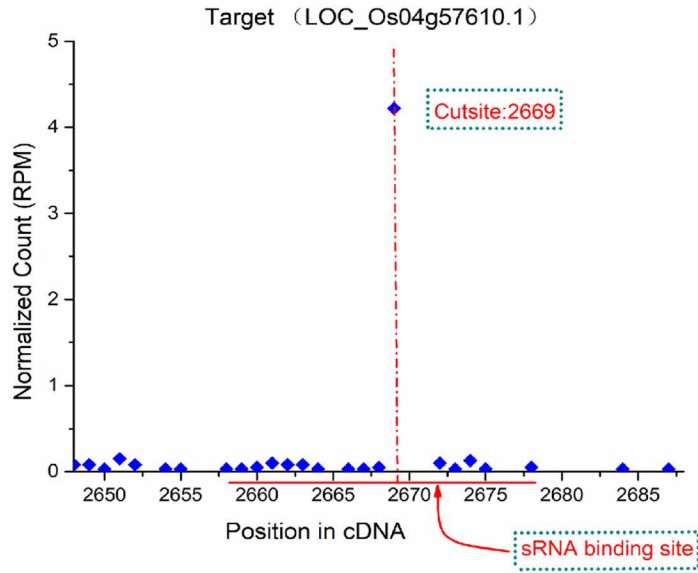

**Fig 4. osa-miR167d-5p and co-regulating siRNAs with their targetARF8 (LOC_Os04g57610) in rice seedling. A**. Expression of osa-miR167d-5p, osa-miR167c-5p and isomiR of osa-miR167d-5p in the samples of control, drought, salt, cold and heat, and their alignments with target genes. The red number indicates that the sRNA plays the most important regulatory role. **B**. Degradome sequencing data library GSM434596 was recruited for T-plot profiling. The ID of the target ARF8 is listed on the top of the figure. The *y* axes measure the normalized reads (in RMP, reads per million) of the degradome signals, and the *x* axes represent the position of the cleavage signals on the target transcripts. The binding site of the osa-miR167d-5p, osa-miR167c-5p and isomiR of osa-miR167d-5p on their target was denoted by red horizontal line.The red vertical dotted line indicates the specific cleavage site.

roots in rice [32]. We found in rice seedlings that the isomiR of osa-miR156j-5p also partici-pates in the co-regulation of OsSPL3/OsSPL12, and its expression level is comparable to that of osa-miR156j-5p (S2 Table), indicating that the isomiR of osa-miR156j-5p also plays an important role in the occurrence of adventitious roots.

In addition, we found that some miRNAs and isomiRs regulate target genes under different conditions. For example, seedling under normal conditions or drought, salt, cold treatment, osa-miR172d-3p was involved in the regulation of AP2 genes (LOC_Os03g60430, LOC_Os04g55560, LOC_Os05g03040, LOC_Os07g13170), but under heat treatment or heat + salt treatment, only the isomiR of osa-miR172d-3p was involved in the regulation (S2 Table).

Besides the 30 isomiRs, 12 other co-regulatory siRNAs were also identified by sRNATar-getDigger (S2 and S3 Tables) and passed psRNATarget validation (S6 Table). The expressions of some siRNAs are tissue or condition specific. For example, siRNA_12 is only expressed in panicle, while other co-regulatory siRNAs are only expressed in seedling. In addition, siRNA_3 and siRNA_12 are only expressed in the heat condition.Compared to isomiRs, the expression levels of these siRNAs are relatively low and may play a supporting role in co-regulation.

## Conclusions

In this study, we re-mined the rice "miRNA→Target" database by sRNATargetDigger and found the regulatory relationship between 51 miRNAs and 48 target genes in the original database could not be verified due to the low expression levels of miRNA, poor complemen-tarity between miRNA and target, or no specific cleavage signal detected by degradome in the middle of the miRNA binding site in the targets. These miRNA→target pairs may be incorrect in these tissues or conditions. In addition, a number of miRNA→target regulatory relation-ships missed in the database have been discovered. We also found 86.2% of the target genes were co-regulated by one or more miRNAs\siRNAs. Besides the known miRNAs, 30 isomiRs and 12 siRNAs were identified to be involved in co-regulation, which play important roles in rice response to external auxin regulation, rice blast resistance, adventitious root formation, etc. Some isomiRs even have higher expression levels than miRNAs.

## Supporting information

**S1 Fig. Target plots (t-plots) of forty one target genes which did not find cleavage signals in the middle of the miRNA binding sites by degradome.** All 5 degradome sequencing data libraries (GSM434596, GSM455938, GSM4113551, GSM455939, GSM476257) were recruited for T-plot profiling. The IDs of the target transcripts and the corresponding miRNAs are listed on the top of sub-figures. The x axes measure the positions of the degradome signals along the transcripts, and the y axes measure the degradome signal intensities based on nor-malized counts (in RPM, reads per million).The binding sites of the miRNA on their target transcripts were denoted by red horizontal lines. No specific cleavage signals have been found in the middle of the miRNA binding sites in these target genes.
(PDF)

**S1 Table. Sequencing library information of sRNA-seq and degradome data for rice miRNA-target re-mining.**
(XLS)

**S2 Table. miRNAs and co-regulatory siRNAs identified in seedling.**
(XLS)

**S3 Table. miRNAs and co-regulatory siRNAs identified in panicle.**
(XLS)

**S4 Table. miRNA-target pairs are not found by sRNATargetDigger.**
(XLS)

**S5 Table. Validation of the Newly discovered miRNA-target pairs in rice by psRNATarget.**
(XLS)

**S6 Table. Validation of the co-regulatory siRNA-target pairs by psRNATarget.**
(XLSX)

## Acknowledgments

We would like to thank all the publicly available data sets and the scientists behind them.

## Author contributions

**Conceptualization:** Chaogang Shao.

**Data curation:** Zhihong Yang, Chaogang Shao.

**Formal analysis:** Zhihong Yang, Yeqing Jiang, Chaogang Shao.

**Funding acquisition:** Chaogang Shao.

**Investigation:** Zhihong Yang, Lan Yu, Yeqing Jiang, Chaogang Shao.

**Methodology:** Lan Yu, Chaogang Shao.

**Project administration:** Chaogang Shao.

**Resources:** Chaogang Shao.

**Software:** Yijun Meng, Chaogang Shao.

**Supervision:** Chaogang Shao.

**Validation:** Chaogang Shao.

**Visualization:** Yijun Meng, Chaogang Shao.

**Writing – original draft:** Zhihong Yang, Chaogang Shao.

**Writing – review & editing:** Yijun Meng, Chaogang Shao.

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
