## [Decision Letter · Decision Letter 0]

17 Dec 2024

PONE-D-24-47896Identification of the co-regulatory siRNAs of “miRNA→Target” in Oryza sativaPLOS ONE

Dear Dr. yang,

Thank you for submitting your manuscript to PLOS ONE. After careful consideration, we feel that it has merit but does not fully meet PLOS ONE’s publication criteria as it currently stands. Therefore, we invite you to submit a revised version of the manuscript that addresses the points raised during the review process.

We look forward to receiving your revised manuscript.

Kind regards,

Keqiang Wu, Ph.D

Academic Editor

PLOS ONE

Journal Requirements:

2. Thank you for stating the following financial disclosure: [This work was supported by the National Natural Sciences Foundation of China[31771457]].

3. Thank you for uploading your study's underlying data set. Unfortunately, the repository you have noted in your Data Availability statement does not qualify as an acceptable data repository according to PLOS's standards.

4. In the online submission form, you indicated that [The datasets used and/or analysed during the current study are available from the corresponding author on reasonable request].

5. Please include captions for your Supporting Information files at the end of your manuscript, and update any in-text citations to match accordingly. Please see our Supporting Information guidelines for more information: http://journals.plos.org/plosone/s/supporting-information .

6. We notice that your figures are uploaded with the file type Supplementary. Please amend the file type to 'Supporting Information'. Please ensure that each Supporting Information file has a legend listed in the manuscript after the references list.

Additional Editor Comments:

Thank you for submitting your manuscript to PLOS ONE. We have now received reports from two reviewers. Please address the comments from both reviewers.

Reviewers' comments:

Reviewer's Responses to Questions

**Comments to the Author**

1. Is the manuscript technically sound, and do the data support the conclusions?

Reviewer #1: Yes

Reviewer #2: Yes

2. Has the statistical analysis been performed appropriately and rigorously? 

Reviewer #1: Yes

Reviewer #2: N/A

3. Have the authors made all data underlying the findings in their manuscript fully available?

Reviewer #1: Yes

Reviewer #2: Yes

4. Is the manuscript presented in an intelligible fashion and written in standard English?

Reviewer #1: Yes

Reviewer #2: Yes

5. Review Comments to the Author

Reviewer #1: In the manuscript entitled " Identification of the co-regulatory siRNAs of “miRNA→Target” in Oryza sativa”, Yang et al. described the use of their previously developed sRNATargetDigger tool and public databases to re-mine the reported rice "miRNA→Target" database, and search for missing co-regulatory siRNAs information. As a result, they found 86.2% of the target genes were co-regulated by one or more miRNAs\siRNAs and some errors in the rice "miRNA→Target" database were also be corrected by authors.

MicroRNAs are the key factors in post-transcriptional expression regulation of genes and are widely involved in the regulation of important agronomic traits in rice. This study improves the rice "miRNA→target" database which is significance for basic research and molecular breeding of rice . So, I recommend Plos one can accept this paper.

However, I have a few comments that I think need to be address.

Comments:

1. The co-regulatory siRNA_2 (“UUUGGAUUGAAGGGAGCUCUGA”) shown in Supplementary Table S2 has one more “A” at the 3' end than the osa-miR159a.1 (“UUUGGAUUGAAGGGAGCUCUG”), which may be due to 3' addition events in miRNA maturation processes and should theoretically be classified as an isomiR of osa-miR159a.1. Authors need to rename all these co-regulatory siRNAs and revise the relevant content in the manuscript.

2. Many miRNA target genes in the manuscript only have id numbers. It would be better to provide gene annotation information to facilitate readers' reading and understanding of miRNA functions.

3. In order to compare different samples, the author normalized the high-throughput sequencing data. However, I found that in Fig. 3, the two degradome(GSM4113551 and GSM434596) detected the same target gene(LOC_Os02g36924.1), but the difference in cleavage signal intensity was several dozen times. Does this difference have an impact on the results?

Reviewer #2: In the manuscript “Identification of the co-regulatory siRNAs of ‘miRNA→Target’ in Oryza sativa”, Yang et al. have made a significant contribution by re-mining the rice "miRNA→Target" database using the sRNATargetDigger tool. Approximately 86.2% of rice target genes were discovered to be co-regulated by more than one miRNA/siRNA. Additionally, 23 isomiRNA and 19 siRNA were identified, some of which are expressed at even higher levels than miRNAs and may be involved in various aspects of biological processes. This work represents an update to the understanding of rice small RNA society and provides a more comprehensive and reliable "miRNA→Target" regulatory information for rice research. The text is well-written and easy to understand. However, some minor revisions are recommended before publication.

1. By utilizing sRNATargetDigger and public data, the authors have uncovered many miRNA-Target pairs that were previously overlooked in the rice "miRNA-Target" database. As far as I am aware, in addition to the TarDB database mentioned in the paper, there are numerous other research papers on the mining of miRNA target genes in rice. It is suggested that the authors explore relevant literature to determine if their results (or partial results) can be supported by other research, which would enhance the reliability of their findings.

2. Many abbreviations lack full name information when they first appear. It is recommended to provide the full names, such as trans-acting small interfering RNA (ta-siRNA), etc.

3. In rice, the authors found that the expression of many isomiRs exceeds that of miRNAs. Is this a sequencing artifact or an annotation error of miRNA? Does this phenomenon also occur in other species?

6. PLOS authors have the option to publish the peer review history of their article (what does this mean? ). If published, this will include your full peer review and any attached files.

**Do you want your identity to be public for this peer review?** For information about this choice, including consent withdrawal, please see our Privacy Policy .

Reviewer #1: No

Reviewer #2: No

---

## [Author Response · Author response to Decision Letter 0]

9 Jan 2025

Response to academic editor

We've checked your submission and before we can proceed, we need you to address the following issues:

1. We understand that your data availability statement currently reads as follows: "The sRNA HTS data sets (GSM1081563, GSM1081564, GSM1081565, GSM1409671, GSM1409672, GSM1409673, GSM1409674, GSM1547539, GSM816687, GSM816688, GSM816689, GSM816690, GSM816691, GSM816692, GSM816693, GSM816694, GSM816695, GSM816696, GSM816697, GSM816698, GSM816699, GSM647192, GSM647193, GSM647194, GSM647195, GSM816730, GSM816731, GSM816732, GSM816733, GSM816735, GSM816736, GSM816737, GSM816738, GSM816739, GSM816740, GSM816741, GSM816742, GSM816743, GSM816744) and the degradome sequencing data sets（GSM434596，GSM455938，GSM4113551，GSM455939，GSM476257）of Oryza sativa were retrieved from GEO (Gene Expression Omnibus; http://www.ncbi.nlm.nih.gov/geo/). The cDNAs and the gene annotations of Oryza sativa were retrieved from the FTP site of the rice genome annotation project established by the Institute for Genome Research (TIGR rice,

Release7;ftp://ftp.plantbiology.msu.edu/pub/data/Eukaryotic_Projects/o_sativa/annotation_dbs/pseudomolecules/). The miRNA-target database of Oryza sativa were retrieved from the TarDB (http://www.biosequencing.cn/TarDB/download.html) and the miRNAs of Oryza sativa were retrieved from miRBase (release 22.1, https://www.mirbase.org/). The mining software, sRNATargetDigger, can be downloaded from the Project sRNATargetDigger in OSF database (https://osf.io/n7uc9/?view_only=6a5e44b06d5949ea9f02d9d5a8cb79ca)

We note that the link provided for the FTP site of the rice genome annotation project established by the Institute for Genome Research (TIGR rice, Release7; ftp://ftp.plantbiology.msu.edu/pub/data/Eukaryotic_Projects/o_sativa/annotation_dbs/pseudomolecules/) leads to an error. Could you please provide a working link to this source?

Reply: Thank you very much for helping us to find the invalid database link in our manuscript. Due to the data updates, the Rice Genome Annotation Project Team has revised the FTP download site to: https://rice.uga.edu/pub/data/Eukaryotic_Projects/o_sativa/annotation_dbs/pseudomolecules/version_7.0/. Additionally, the OSF database provides a new download address for the sRNATargetDigger software: https://osf.io/n7uc9/files/osfstorage. (The original address is also valid, but the download interface is not as simple.).

We have amended the relevant information in the manuscript and the Data Availability statement has been revised as follows:

The sRNA HTS data sets (GSM1081563, GSM1081564, GSM1081565, GSM1409671, GSM1409672, GSM1409673, GSM1409674, GSM1547539, GSM816687, GSM816688, GSM816689, GSM816690, GSM816691, GSM816692, GSM816693, GSM816694, GSM816695, GSM816696, GSM816697, GSM816698, GSM816699, GSM647192, GSM647193, GSM647194, GSM647195, GSM816730, GSM816731, GSM816732, GSM816733, GSM816735, GSM816736, GSM816737, GSM816738, GSM816739, GSM816740, GSM816741, GSM816742, GSM816743, GSM816744) and the degradome sequencing data sets（GSM434596，GSM455938，GSM4113551，GSM455939，GSM476257）of Oryza sativa were retrieved from GEO (Gene Expression Omnibus; http://www. ncbi.nlm.nih.gov/geo/). The cDNAs and the gene annotations of Oryza sativa were retrieved from the FTP site of the rice genome annotation project (Release7; https://rice.uga.edu/pub/data/Eukaryotic_Projects/o_sativa/annotation_dbs/pseudomolecules/version_7.0/). The miRNA-target database of Oryza sativa were retrieved from the TarDB (http://www.biosequencing.cn/TarDB/download.html) and the miRNAs of Oryza sativa were retrieved from miRBase (release 22.1; https://www.mirbase.org/). The mining software, sRNATargetDigger, can be downloaded from the Project sRNATargetDigger in OSF database (https://osf.io/n7uc9/files/osfstorage).

2. When submitting your revision, we need you to address these additional requirements. Please ensure that your manuscript meets PLOS ONE's style requirements, including those for file naming. The PLOS ONE style templates can be found at https://journals.plos.org/plosone/s/file?id=wjVg/PLOSOne_formatting_sample_main_body.pdf and https://journals.plos.org/plosone/s/file?id=ba62/PLOSOne_formatting_sample_title_authors_affiliations.pdf

Reply: Thank you. We have checked and modified the manuscript to meet the style requirements of PLOS ONE's .

3. Thank you for stating the following financial disclosure: [This work was supported by the National Natural Sciences Foundation of China[31771457]]. Please state what role the funders took in the study. If the funders had no role, please state: ""The funders had no role in study design, data collection and analysis, decision to publish, or preparation of the manuscript."" If this statement is not correct you must amend it as needed. Please include this amended Role of Funder statement in your cover letter; we will change the online submission form on your behalf.

Reply: Thank you. "The funder had no role in study design, data collection and analysis, decision to publish, or preparation of the manuscript." has been added in the Founding information.

4. Please include captions for your Supporting Information files at the end of your manuscript, and update any in text citations to match accordingly. Please see our Supporting Information guidelines for more information: http://journals.plos.org/plosone/s/supporting-information.

Reply:Thank you. The captions of Supporting Information files have been added at the end of the revised manuscript and intext citations have been updated as needed.

5. We notice that your figures are uploaded with the file type Supplementary. Please amend the file type to 'Supporting Information'. Please ensure that each Supporting Information file has a legend listed in the manuscript after the references list.

Reply:Thank you. The file type 'Supplementary' has been amended to 'Supporting Information' and the legends of Supporting Information files have been listed in the revised manuscript after the references list.

Reply: Thank you. The reference list has been checked again. Because of the rice genome annotation database updates, the reference 【19】, “Q Yuan，S Ouyang，L Jia，S Bernard，C Foo，S Razvan, et al. The TIGR rice genome annotation resource: annotating the rice genome and creating resources for plant biologists. Nucleic Acids Res. 2003;31(1):229-233.” has been replaced by “Kawahara Y, de la Bastide M, Hamilton JP, Kanamori H, McCombie WR, Ouyang S , et al. Improvement of the Oryza sativa Nipponbare reference genome using next generation sequence and optical map data. Rice. 2013; 6(1): 4.”

One reference has been added as showing below which was accompanied with revised manuscript.

【25】Yuhan Fei , Rui Wang , Haoyuan Li , Shu Liu, Hongsheng Zhang , Ji Huang . DPMIND: degradome-based plant miRNA-target interaction and network database.Bioinformatics. 2018, 34(9):1618-1620

Response to the Reviewers

Reviewers' comments:

Reviewer #1:

In the manuscript entitled " Identification of the co-regulatory siRNAs of “miRNA→Target” in Oryza sativa”, Yang et al. described the use of their previously developed sRNATargetDigger tool and public databases to re-mine the reported rice "miRNA→Target" database, and search for missing co-regulatory siRNAs information. As a result, they found 86.2% of the target genes were co-regulated by one or more miRNAs\siRNAs and some errors in the rice "miRNA→Target" database were also be corrected by authors.

MicroRNAs are the key factors in post-transcriptional expression regulation of genes and are widely involved in the regulation of important agronomic traits in rice. This study improves the rice "miRNA→target" database which is significance for basic research and molecular breeding of rice . So, I recommend Plos one can accept this paper. However, I have a few comments that I think need to be address.

Comments:

1. The co-regulatory siRNA_2 (“UUUGGAUUGAAGGGAGCUCUGA”) shown in Supplementary Table S2 has one more “A” at the 3' end than the osa-miR159a.1 (“UUUGGAUUGAAGGGAGCUCUG”), which may be due to 3' addition events in miRNA maturation processes and should theoretically be classified as an isomiR of osa-miR159a.1. Authors need to rename all these co-regulatory siRNAs and revise the relevant content in the manuscript.

Reply: Thank you very much for finding some mistakes in our manuscript. We carefully analyzed the 19 co-regulatory siRNAs in the original manuscript and found that 7 of them had high similarity with miRNAs. These siRNA sequences do not completely match the pre-miRNA sequence, which may be due to 3' addition events in miRNA maturation processes or base modification of miRNA. We re-annotated them as isomiRs and adjusted the numbering of the other 12 co-regulatory siRNAs accordingly. The relevant contents have also been revised and marked in red in the manuscript and supplementary tables of S2, S3, and S6.

ID of the co-regulatory siRNAs in the original manuscript ID of the co-regulatory siRNA re-annotated in the revised paper

siRNA_2 isomiR_of_osa-miR159a.1\osa-miR159b_2

siRNA_4 isomiR_of_osa-miR159a.1\osa-miR159b_3

siRNA_6 isomiR_of_osa-miR396e\osa-miR396f_2

siRNA_7 isomiR_of_osa-miR166a-3p\osa-miR166b-3p\osa-miR166c-3p\osa-miR166d-3p\osa-miR166f\osa-miR166j-3p

siRNA_8 isomiR_of_osa-miR820a\osa-miR820b\osa-miR820c_4

siRNA_12 isomiR_of_osa-miR820a\osa-miR820b\osa-miR820c_5

siRNA_19 isomiR_of_osa-miR408-3p

siRNA_1 siRNA_1

siRNA_3 siRNA_2

siRNA_5 siRNA_3

siRNA_9 siRNA_4

siRNA_10 siRNA_5

siRNA_11 siRNA_6

siRNA_13 siRNA_7

siRNA_14 siRNA_8

siRNA_15 siRNA_9

siRNA_16 siRNA_10

siRNA_17 siRNA_11

siRNA_18 siRNA_12

2. Many miRNA target genes in the manuscript only have id numbers. It would be better to provide gene annotation information to facilitate readers' reading and understanding of miRNA functions.

Reply: Thank you very much for your constructive comments. We have added the annotation information to all target genes of miRNAs in the manuscript and marked them in red.

3. In order to compare different samples, the author normalized the high-throughput sequencing data. However, I found that in Fig. 3, the two degradome(GSM4113551 and GSM434596) detected the same target gene(LOC_Os02g36924.1), but the difference in cleavage signal intensity was several dozen times. Does this difference have an impact on the results?

Reply: Thank you. We found that normalization is effective for comparing high-throughput sequencing data of different samples from the same experiment, but it does have some differences for the data generated by different laboratories or different experimental methods. Therefore, the high-throughput sequencing data are separately input into sRNATargetDigger for mining to prevent omissions. In addition, the signal intensity measured by the two degradomes of GSM4113551 and GSM434596 did indeed differ by several times, but in each T-plot, the specific cleavage signal could clearly distinguish from the background signal, and the results were consistent.

Reviewer #2: In the manuscript “Identification of the co-regulatory siRNAs of ‘miRNA→Target’ in Oryza sativa”, Yang et al. have made a significant contribution by re-mining the rice "miRNA→Target" database using the sRNATargetDigger tool. Approximately 86.2% of rice target genes were discovered to be co-regulated by more than one miRNA/siRNA. Additionally, 23 isomiRNA and 19 siRNA were identified, some of which are expressed at even higher levels than miRNAs and may be involved in various aspects of biological processes. This work represents an update to the understanding of rice small RNA society and provides a more comprehensive and reliable "miRNA→Target" regulatory information for rice research. The text is well-written and easy to understand. However, some minor revisions are recommended before publication.

1. By utilizing sRNATargetDigger and public data, the authors have uncovered many miRNA-Target pairs that were previously overlooked in the rice "miRNA-Target" database. As far as I am aware, in addition to the TarDB database mentioned in the paper, there are numerous other research papers on the mining of miRNA target genes in rice. It is suggested that the authors explore relevant literature to determine if their results (or partial results) can be supported by other research, which would enhance the reliability of their findings.

Reply: Thank you very much for your constructive comments. We investigated the relevant literature on the mining of miRNA target genes in rice and found our partial results, the regulatory relationships of "osa-miR444c.1→LOC_Os02g36924.1 (OsMADS27)", "osa-miR444c.2→LOC_Os02g49840.2 (OsMADS57)", "osa-miR444e→LOC_Os02g36924.1 (OsMADS27)" and "osa-miR444e→ LOC_Os04g38780.1(transcription factor)" were also supported by the research of Fei et al.[1] . This information has been added into the revised manuscript.

[1] Fei et al. DPMIND: degradome-based plant miRNA-target interaction and network database.Bioinformatics. 2018, 34(9):1618-1620

2. Many abbreviations lack full name information when they first appear. It is recommended to provide the full names, such as trans-acting small interfering RNA (ta-siRNA), etc.

Reply: Thank you very much for finding some writing defects in our manuscript. The full names have been added to the abbreviations lacking these information and marked in red in the manuscript.

3. In rice, the authors found that the expression of many isomiRs exceeds that of miRNAs. Is this a sequencing artifact or an annotation error of miRNA? Does this phenomenon also occur in other species?

Reply: Thank you. High-throughput sequencing revealed that many miRNAs have a large number of isomiRs. The current research shows that these isomiRs are not sequencing artifact. Phillipe Loher et al. have sequenced five samples a total of seven times each in different sequencing centers and found that the isomiRs expression profile was highly consistent among the replicate samples, ruling out the possibility of sequencing errors and random degradation [1]. Other researchers also have found some isomiRs with expression values exceeding miRNA in different species, such as 5’isomiR of hsa-miR 140-3p in human[2], isomiRs of miR528-5p, miR171, miR156, miR408-5p, miR1320-5p in rice[3], and isomiR of miR159c in somatic embryo and seedling of larch[4]. Currently, a large number of isomiRs with regulatory functions are being discovered. Some 5’isomiRs even possess seed sequences different from those of miRNAs, enabling them to regulate different target genes[2]. Therefore, many scholars believe that the annotation of miRNA as a single sequence in miRBase may not reflect its true biological function, and suggest that information on isomiRs should also be integrated into the miRNA database[1, 2, 5].

[1] Phillipe Loher et al. IsomiR expression profiles in human lymphoblastoid cell lines exhibit population and gender dependencies. Oncotarget. 2014, 5(18):8790-8802

[2] Omar Salem et al. The highly expressed 5'isomiR of hsa-miR-140-3p contributes to the tumor-suppressive effects of miR-140 by reducing breast cancer proliferation and migration. BMC Genomics. 2016 , 17:566.

[3] Sonia Balyan et al.Investigation into the miRNA/5' isomiRNAs function and drought-mediated miRNA processing in rice.Functional & Integrative Genomics,2020, 20:509–522

[4]Junhong Zhang et al. A genome-wide survey of microRNA truncation and 30 nucleotide addition events in larch (Larix leptolepis).Planta, 2013, 237:1047–1056

[5] Desvignes T et al. MiRNA nomenclature: a view incorporating genetic origins, biosynthetic pathways, and sequence variants. Trends Genet, 2015, 31:613–626

---

## [Decision Letter · Decision Letter 1]

3 Mar 2025

Identification of the co-regulatory siRNAs of “miRNA→Target” in Oryza sativa

PONE-D-24-47896R1

Dear Dr. Shao,

We’re pleased to inform you that your manuscript has been judged scientifically suitable for publication and will be formally accepted for publication once it meets all outstanding technical requirements.

Kind regards,

Keqiang Wu, Ph.D

Academic Editor

PLOS ONE

Additional Editor Comments (optional):

Reviewers' comments:

Reviewer's Responses to Questions

**Comments to the Author**

1. If the authors have adequately addressed your comments raised in a previous round of review and you feel that this manuscript is now acceptable for publication, you may indicate that here to bypass the “Comments to the Author” section, enter your conflict of interest statement in the “Confidential to Editor” section, and submit your "Accept" recommendation.

Reviewer #2: All comments have been addressed

2. Is the manuscript technically sound, and do the data support the conclusions?

Reviewer #2: Yes

3. Has the statistical analysis been performed appropriately and rigorously? 

Reviewer #2: N/A

4. Have the authors made all data underlying the findings in their manuscript fully available?

Reviewer #2: Yes

5. Is the manuscript presented in an intelligible fashion and written in standard English?

Reviewer #2: Yes

6. Review Comments to the Author

Reviewer #2: (No Response)

7. PLOS authors have the option to publish the peer review history of their article (what does this mean? ). If published, this will include your full peer review and any attached files.

**Do you want your identity to be public for this peer review?** For information about this choice, including consent withdrawal, please see our Privacy Policy .

Reviewer #2: **Yes: ** Ming Chen

---

## [Editor Report · Acceptance letter]

PONE-D-24-47896R1

PLOS ONE

Dear Dr. Shao,

I'm pleased to inform you that your manuscript has been deemed suitable for publication in PLOS ONE. Congratulations! Your manuscript is now being handed over to our production team.

Kind regards,

on behalf of

Professor Keqiang Wu

Academic Editor

PLOS ONE